# Lyophilized Human Bone Allograft as an Antibiotic Carrier: An In Vitro and In Vivo Study

**DOI:** 10.3390/antibiotics11070969

**Published:** 2022-07-19

**Authors:** Débora C. Coraça-Huber, Stephan J. M. Steixner, Stevo Najman, Sanja Stojanovic, Ronja Finze, Denis Rimashevskiy, Dina Saginova, Mike Barbeck, Reinhard Schnettler

**Affiliations:** 1Research Laboratory for Biofilms and Implant Associated Infections (BIOFILM LAB), Experimental Orthopaedics, University Hospital for Orthopaedics and Traumatology, Medical University of Innsbruck, Peter-Mayr-Strasse 4b, Room 204, 6020 Innsbruck, Austria; stephan.steixner@i-med.ac.at; 2Department of Cell and Tissue Engineering, Scientific Research Center for Biomedicine, Faculty of Medicine, University of Niš, Blvd. Dr Zorana Djindjica, 18108 Niš, Serbia; stevo.najman@gmail.com (S.N.); s.sanja88@gmail.com (S.S.); 3Department of Biology and Human Genetics, Faculty of Medicine, University of Niš, Blvd. Dr Zorana Djindjica, 18108 Niš, Serbia; 4University Medical Centre, Justus Liebig University of Giessen, 35390 Giessen, Germany; ronja.finze@freenet.de (R.F.); reiner.schnettler@mac.com (R.S.); 5Department of Traumatology and Orthopedics, Peoples Friendship University of Russia, Miklukho-Maklaya Street 6, 117198 Moscow, Russia; drimashe@gmail.com; 6National Scientific Center of Traumatology and Orthopedics Named after Academician N. D. Batpenov, 15a Abylay khan Ave., Nur-Sultan 01000, Kazakhstan; sa_dina@mail.ru; 7Clinic and Policlinic for Dermatology and Venereology, University Medical Center Rostock, 18057 Rostock, Germany; mike.barbeck@med.uni-rostock.de; 8BerlinAnalytix GmbH, 12109 Berlin, Germany

**Keywords:** allograft, drug delivery, local antibiotic delivery, rabbit tibia, MIC90, bone graft infection, antibiotic susceptibility

## Abstract

**Background**: Antibiotics delivered from implanted bone substitute materials (BSM) can potentially be used to prevent acute infections and biofilm formation, providing high concentrations of antibiotics at the surgical site without systemic toxicity. In addition, BSM should allow osteoconductivity supporting bone healing without further surgery. Promising results have been achieved using lyophilized bone allografts mixed with antibiotics. **Methods**: In this study specially prepared human bone allografts were evaluated as an antibiotic carrier in vitro and in vivo. The efficacy of different antibiotic-impregnated bone allografts was measured by drug release tests in vitro and in vivo and bacterial susceptibility tests using four bacterial species usually responsible for implant-associated infections. **Results**: The loading procedures of allograft bone substitutes with antibiotics were successful. Some of the antibiotic concentrations exceeded the MIC90 for up to 7 days in vitro and for up to 72 h in vivo. The susceptibility tests showed that *S. epidermidis* ATCC 12228 was the most susceptible bacterial species in comparison to the other strains tested for all antibiotic substances. Vancomycin and rifampicin showed the best results against standard and patient-isolated strains in vitro. In vivo, new bone formation was comparable in all study groups including the control group without antibiotic loading. **Conclusions**: Human bone allografts showed the capacity to act as customized loaded antibiotic carriers to prevent acute infections and should be considered in the management of bone infections in combination with systemic antimicrobial therapy.

## 1. Introduction

Bone substitute materials (BSM) are an important tool to reconstruct bone functionality in different medical disciplines. Preserved bones are commonly used in prosthesis allograft composites. Used for many decades, the method involves structural allografts developed for bone replacement in the course of revision hip replacement surgery [1,2,3] or in knee arthroplasty [4]. These are typically complex clinical cases, where the patient has already undergone several operations and the revision is the last attempt for creating a moving, weight-bearing joint. Despite the high rate of complications largely due to the reduced remodeling of the grafts, the clinical results are usually better with grafts than without [4,5,6]. However, bone allografts bear a higher risk of transmission of diseases and local contamination compared with processed bone grafts [7,8].

Surgery with bone allografts is complex and time-consuming; therefore, it is also per se prone to a higher infection rate (2.0% to 2.5%) [9,10,11]. The prevalence of implant-associated infections varies between 0.5% and 20% depending on the location of the implant [12,13,14]. Infection rates are particularly high in dental surgery or after open fractures [12,13,14] and approximately 77.3% of infection incidents required surgical intervention as systematic administration of antibiotics was not sufficient [15]. Additionally, the impaction used for placing bone transplants can disrupt the local circulation and reduce bone ingrowth [16,17,18,19]. In the case of a site infection, systemically administered antibiotics cannot reach the infected bone graft [20]. *Staphylococcus epidermidis* and *Staphylococcus aureus* are the strains, which mostly colonize orthopaedic implant surfaces [21]. The main reason for treatment failure is the formation of biofilms, which enables bacteria within the biofilm to become less susceptible to antibiotic treatment [22,23,24].

Biomaterials loaded with antibiotics are frequently used in orthopaedic surgeries [25], but also applied in oral and maxillofacial surgery [26,27]. When combined with systemic antibiotic treatment, this form of application can offer advantages. Such advantages are related to the local administration which achieves higher antibiotic concentrations at the site of infection without causing systemic side effects and thereby reduces the development of resistances [26,28], and; to its prophylactic approach that reduces the chances of biofilm formation and consequent antibiotic resistance [29]. Currently, bone cements impregnated with antibiotics are frequently used in orthopaedic surgery, but new materials including bone allografts are also targets of several studies [27,30].

Due to the excellent osteoconductive properties, mineralized allogenic BSM are an ideal biomaterial to regenerate large bone defects and are more frequently investigated for application in orthopaedic, trauma, or dental surgery [30,31]. Antibiotic loading of allogenic BSM to reduce osteomyelitis is present in several studies in vitro and in vivo [32,33,34,35]. It was shown that several antibiotics, e.g., aminoglycosides, β-lactams, lincosamides, glycopeptides, ansamycines and fluorchinolones, adhered sufficiently to the allogeneic bone tissue and were released slowly [30]. The results also showed that high concentrations of locally administered antibiotics selectively promoted growth and colonization of tissue-characteristic cells such as osteoblasts on the BSM surfaces and did not interfere with the bone regeneration process in vitro [26,27,28,30]. Nevertheless, it is unknown for which period locally delivered antibiotics exceed minimal inhibitory concentrations in vivo and how can such material efficiently prevent infections.

Therefore, the aim of the present study was to evaluate the efficacy of human bone allografts prepared according to the C+TBA processing (Cell+Tissuebank Austria GmbH, Krems, Austria) to act as an antibiotic carrier. The capacity of the allografts impregnated with different antibiotics was measured by drug release tests in vitro and in vivo, as well as bacterial susceptibility using four different bacterial species. Finally, new bone formation was additionally determined in vivo at 4 months post-implantation using established histological and histomorphometrical analysis methods [36].

## 2. Materials and Methods

### 2.1. Human Bone Allografts

In this study, we used lyophilized human bone allografts cubes (4 mm edge length). Samples were prepared according to the C+TBA process (Cell+Tissuebank Austria GmbH, Krems, Austria). The C-TBA process for the cleaning and sterilization of the bone allografts was carried out as follows: after shaping and crude cleaning, the donor tissue undergoes ultra-sonication to remove blood, cells, and tissue components, mainly for the removal of fat from the cancellous structure. In a chemical treatment, all non-collagenic proteins are denatured, potential viruses are inactivated, and bacteria are destroyed. In the subsequent oxidative treatment, persisting soluble proteins are denatured and potential antigenicity is eliminated. Finally, the allograft tissue undergoes lyophilization, a dehydration technique. The final sterilization by gamma irradiation guarantees a sterility assurance level (SAL) of 10^6^ while ensuring the structural and functional integrity of the product and its packaging.

### 2.2. Antibiotic Substances

Gentamicin sulphate total load 80 mg/mL (Gentamicin-ratiopharm, ratiopharm GmbH, Ulm, Germany), vancomycin hydrochloride total load 50 mg/mL (Vancomycin Lyomark, Lyomark Pharma GmbH, Oberhaching, Germany), clindamycin phosphate total load 150 mg/mL (Clindamycin Kabi 150 mg/mL, Fresenius Kabi Deutschland GmbH, Bad Homburg, Germany), rifampicin (EREMFAT^®^, RIEMSER Pharma GmbH, Greifswald, Germany) total load 60 mg/mL. The mixture of rifampicin and vancomycin used a total load of 110 mg/mL (1:1 mixture).

### 2.3. Antibiotic Impregnation of the Bone Allografts

For the impregnation of the bone allograft cubes, each antibiotic substance was diluted in water for injection (Fresenius Kabi Deutschland GmbH, Bad Homburg, Germany) following the manufacturer procedures to obtain the concentration above mentioned. One piece of each bone allograft cube was immersed in 2 mL of each antibiotic substance and stored at room temperature for 10 min. After 10 min the impregnated samples were removed from the antibiotic solutions and immediately used for the in vitro and the in vivo tests. The experiments were carried out in triplicates.

### 2.4. In Vitro Analysis

#### 2.4.1. Antibiotic Release Kinetic Measurements

Impregnated allografts were immersed in 2 mL phosphate-buffered saline (PBS, pH 7.4, Sigma-Aldrich, Schnelldorf, Germany), vortexed for 1 min, and incubated at 37 °C. After 1, 2, 3, 4, 5, 6, and 7 days, the elusion was completely removed and stored at −20 °C until release kinetic measurement. Fresh PBS was added for the next measurement. Release kinetic measurement was performed using a conventional microbiological agar diffusion assay with *Bacillus subtilis* (in Test Agar pH 8.0, Merck KGaA, Darmstadt, Germany). A total of 100 µL of each elusion or 10-fold dilutions (from 10.000 to 0.01 mg/L) of each antibiotic was added in a central hole (diameter: 6 mm). Plates were incubated at 37 °C under aerobic conditions. After 24 h, the inhibition zone was measured for each plate (in cm). A standard curve was obtained by logarithmic regression and used to predict the concentration of each antibiotic for each elution. The experiments were carried out in triplicates.

#### 2.4.2. Susceptibility Tests

10 µL of a suspension of methicillin-sensitive *Staphylococcus aureus* ATCC 29212 (MSSA), Methicillin-resistant *Staphylococcus aureus* (MRSA) patient isolated strain, *Staphylococcus epidermidis* ATCC 12228, and *Staphylococcus epidermidis* patient isolated strain suspensions with a concentration of 2 × 10^5^ CFU/mL (0.5 McFarland) were evenly distributed on a Müller-Hinton agar plate (Merck KGaA, Darmstadt, Germany). The clinical isolates were harvested from patients undergoing periprosthetic joint infection (PJI) treatment. 100 µL of each antibiotic elution was added in a central hole (diameter: 6 mm). Plates were incubated at 37 °C for 24 h under aerobic conditions. After incubation, the zones of inhibition were measured on each plate and concentrations were calculated according to the standard curve obtained in the previous section. The experiments were carried out in triplicates.

### 2.5. In Vivo Analysis

#### 2.5.1. Animal Model

In vivo experiments were conducted on 32 (*n* = 4 per group and time point) male rabbits (New Zealand White), obtained from Military Medical Academy, Serbia, with an age of 4–5 months and an average weight of 3 to 3.5 kg. Animals were randomly assigned to the study groups. The experiments were conducted at the Faculty of Medicine (University of Niš, Serbia) and all animals were kept under standard housing conditions (12/12 h light/dark cycle) with unlimited access to standard food and tap water. The animal experiments were authorized by the local Ethical Committee of the Faculty of Medicine (University of Niš, Serbia) based on the approval of the Veterinary Directorate of the Ministry of Agriculture, Forestry and Water Management of the Republic of Serbia (number of approval: 323-07-00278/2017-05/4). The surgical procedure was performed under general anesthesia achieved by intramuscular injection of the combination of ketamine (25–50 mg/kg) and xylazine (2–5 mg/kg). The operation field was shaved, disinfected, and covered with a sterile foil. After an incision of approx. 15 mm in the tibia at the level of the medial condyle, a drilling hole with a diameter of 4 mm and a length of 4 mm was established in the direction of the lateral condyle. Allograft bone blocks were inserted in the tibia and then covered with barrier membranes (Jason^®^, botiss GmbH, Berlin, Germany) to fix the defect site and the allograft cubes within the implant site if needed. Muscle, fascia, and skin were closed using resorbable suture material. One impregnated allograft bone block was implanted in the left and one in the right tibia. In the control group, each tibia defect on both sites was filled with an allograft bone block not soaked with antibiotics and served to compare bony ingrowth. The animals were monitored on a regular basis and as an analgesic treatment all animals received ketoprofen (2 mg/kg) postoperatively, immediately after surgery, and day after surgery. After 1, 3, and 120 days, the animals were sacrificed by lethal doses of barbiturate pentobarbital (Euthasol^®^ 400 mg/mL, GENERA, Croatia) applied intravascularly according to the guidelines for animal euthanasia. Tibias were dissected, allograft bone blocks were removed for kinetic measurements or the whole tibias were obtained for histologic analysis of bone repair (Figure 1).

#### 2.5.2. Determination of the Residual Antibiotic Concentration in Bone Allograft Blocks In Vivo

Bone allograft cubes were removed from the implanted site. After 1 and 3 days but also after 40 and 120 days the bone cubes could be manually separated from the surrounding tissue after careful preparation of the surrounding (bone) tissue by means of a (autopsy) bone saw (A1 Medical, Laurel, MS, USA). After removal of the allograft bone blocks, each allograft cube was placed centrally in *Bacillus subtilis* (Merck KGaA, Germany) in test agar pH 8.0 (Merck KGaA, Germany) inoculated agar plates and the test was performed as described in Section 2.4.1.

#### 2.5.3. Histopathological and Histomorphometrical Evaluation of Bone Regeneration

The histological procedures and the histopathological and histomorphometrical analyses were conducted as previously described [37,38,39,40].

Briefly, the animals were euthanized with an overdose of the above-described anaesthetic after the course of the experiments. The implanted bone blocks and the surrounding tissue were explanted and immediately fixed via a 4% formalin solution for 24 h. Afterward, the explants were dehydrated via a series of increasing alcohol concentrations, and a final xylol exposure was performed. The following decalcification using 10% ethylenediaminetetraacetic acid (Fluka, Schwerte, Germany) at room temperature for 7–10 days was conducted before paraffin embedding and sectioning using a rotation microtome (Leica, Wetzlar, Germany) that resulted in 3–5 μm thick sections. These slides were then stained with haematoxylin and eosin (H&E). The histopathological analysis focused on the interaction of the bone graft blocks (and the added antibiotics) with the surrounding tissue and its osteointegration. For this analysis, we used a light microscope Axio Scope.A1 (Carl Zeiss Microscopy GmbH, Munich, Germany), and an established protocol was applied as previously published [37,38,39,40].

In brief, the qualitative histological evaluation included the observation of the cells participating in the process of biomaterial integration and the possible adverse reactions such as fibrotic encapsulation or necrosis. Histological figures were taken by a microscope camera (Nikon DS-Fi1, Tokyo, Japan) that was connected to an acquisition unit (Nikon digital sight control unit, Tokyo, Japan). The histomorphometrical analyses included the comparative measurements of the tissue fractions within the implant beds of the bone blocks, i.e., the amounts of newly formed bone, remaining bone substitute material, and connective tissue, on basis of previously published methods [37,39,41].

Briefly, the so-called “total scans” were generated by means of a specialized scanning microscope (M8 digital microscope, PreciPoint GmbH, Freising, Germany) connected to a PC system running the PreciPoint software with a 100× magnification in a resolution of 2500 × 1200 pixels. The final total scans contained the complete implant area of the bone blocks as well as the peri-implant tissue. To measure the different tissue fractions within the bone defects, the complete area of the defect sides was first calculated with the “area tool” (in mm^2^) of the PreciPoint software. Afterward, the different fractions were manually measured (in mm^2^) so that finally the fractions were determined by calculating in relation to the total implant area (in %).

### 2.6. Statistical Analysis

The design of the graphics and the statistical analysis were carried out using GraphPad Prism^®^ (Version 8.1.3, GraphPad Software, Inc., San Diego, CA, USA). In vitro tests were carried out in technical triplicates and the statistical evaluation was performed using two-way ANOVA and post hoc Tukey multiple comparisons test. A *p*-value ≤ 0.05 was considered statistically significant.

## 3. Results

### 3.1. In Vitro Antibiotic Release Kinetics

The concentrations of vancomycin and the combination of vancomycin and rifampicin remained above the MIC90 whereas the concentrations of clindamycin and gentamicin fell below the MIC90 on days three and six respectively (Figure 2). Significant differences (*p* ≤ 0.05) of the clindamycin concentration were detected comparing its initial concentration on day one to day seven; the concentration on day one to day two until day seven and the concentration on day two to the concentration on day three to seven (Figure 2A). Significant differences (*p* ≤ 0.05) in the vancomycin concentration were detected comparing the initial concentration to the concentration on day one to day seven, the concentration on day one to day two to seven, and the concentration on day two to the concentration on day three to seven (Figure 2B). The release of gentamicin showed significant differences (*p* ≤ 0.05) comparing the initial concentration to the concentration on days one to seven and the concentration on day five to the concentrations on days six and seven (Figure 2C). The combination of vancomycin and rifampicin showed significant differences (*p* ≤ 0.05) comparing the initial concentration to the concentrations on day one to day seven, the concentration on day one to the concentrations on day two to seven, the concentration on day two to the load on day three to seven, the load on day three to the concentrations on day four to seven as well as comparing the concentration on day five to the day six and seven (Figure 2D).

### 3.2. Susceptibility Tests

The susceptibility tests showed that *S. epidermidis* ATCC 12228 was the most susceptible strain in comparison to *S. aureus* ATCC 29213 and the clinical isolates of MRSA and *S. epidermidis* (Figure 3). Clindamycin showed a lower activity against *S. epidermidis* ATCC 12228 in comparison to *S. aureus* ATCC 29213 and a very low to zero activity against the clinical isolates of MRSA and *S. epidermidis*. In comparison to all antibiotics, clindamycin showed the least activity against all tested bacterial strains and fell below the MIC90 after the third day of incubation (Figure 3A). The released amount of vancomycin showed a similar activity against *S. epidermidis* ATCC12228, *S. aureus* 29213 and *S. epidermidis* clinical isolate, but a lower efficiency against MRSA (Figure 3B). Gentamicin showed higher activity against *S. epidermidis* ATCC 12228, *S. aureus* ATCC 29213, and MRSA compared to vancomycin and clindamycin. Released gentamicin showed the weakest activity against *S. epidermidis* clinical isolate (Figure 3C). Vancomycin and rifampicin showed high activity against *S. epidermidis* ATCC 12228, *S. aureus* ATCC 29213 and MRSA but a medium activity against the clinical isolate of *S. epidermidis*, similar to vancomycin. The mixture of vancomycin and rifampicin showed the best results against the standard and patient-isolated strains in vitro (Figure 3D).

### 3.3. In Vivo Antibiotic Release Kinetics

The initial concentrations of all antibiotics were significantly reduced on day 1 and 3 post-implantation (*p* ≤ 0.0001) (Figure 4). Moreover, none of the prior antibiotic load could be detected at day 120 post-implantation in all study groups. The concentrations of clindamycin (Figure 4A), vancomycin (Figure 4B), and the combination of vancomycin and rifampicin (Figure 4D) remained above the MIC90 up to 3 days post-implantation, whereas the concentrations of gentamicin fell below the MIC90 at day one and three respectively (Figure 4C).

### 3.4. Histopathological Evaluation of the Tissue Reactions to the Allografts

The histopathological analysis showed that the implanted allogeneic bone blocks were detectable within their implantation beds at every time point, i.e., at 1-, 3- and 120-days post-implantation, in every group (Figure 5). After days 1 and 3, microscopic remnants of the different antibiotics were detectable within the intertrabecular spaces of the bone blocks (data not shown). Furthermore, a fibrin network was visible, in combination with a cell infiltrate composed of different blood cells comparable to a blood clot in all study groups including the control group (data not shown). At these early post-implant time points no ingrowth of complex tissue structures such as collagen or blood vessels was observed. Four months post-implantation, a comparable histological appearance was detected within all study groups including the control group. Newly formed bone covered most of the surfaces of the allogeneic bone blocks (Figure 5A,B). Furthermore, active bone growth indicated by osteoblast accumulations was observable (Figure 5C). At the surface areas of the allogeneic bone blocks that were adherent to the surrounding connective tissue, most often multinucleated giant cells (MNGCs) were observed in combination with clearly visible resorption lacunae (Figure 5B,D,E). The MNGCs that were adherent to the bone blocks were adjacent to multinucleated cells located at the surface of the newly formed bone matrix (Figure 5E). Within the sur-rounding connective tissue low histological signs of inflammatory processes were observable in all study groups indicated by low numbers of macrophages and lymphocytes (Figure 5D). No signs of material rejections and no histological signs of the former antibiotic loadings were found at this late post-implantation timepoint.

### 3.5. Histomorphometrical Analysis of Bone Ingrowth

The results of the histomorphometrical measurements of the tissue distribution within the implant beds of the different study groups revealed comparable amounts of newly formed bone tissue in all five study groups at 4-months’ time point post implantation (Figure 6 and Table 1). Thus, 37.84 ± 6.33% of newly formed bone in the control group, 28.28 ± 7.81% in the vancomycin group, 21.67 ± 8.19% in the clindamycin group, 35.86 ± 8.65% in the gentamicin group and 26.81 ± 6.92% in the group of the combination of vancomycin and rifampicin were detected.

## 4. Discussion

In this study lyophilized bone allograft blocks were immersed in different antibiotic solutions to study the antibiotic release kinetics in vitro and in vivo and the effect of the antibiotic loaded bone blocks on bone remodeling in vivo.

This method is an efficient and easily applicable method for bone tissue loading with antibiotics since the tissue would act as a sponge absorbing the solution. According to Sorger et al., who impregnated grafts for up to 100 h in an antibiotic solution, a long-time incubation might influence the mechanical stability of the bone [42]. Based on Parrish, Witsø and collaborators, mechanical testing of osteochondral and structural allografts impregnated with antibiotics in solutions should be performed before this option is taken into clinical use [43]. In the present study, allogenic bone blocks were immersed for only 10 min in an antibiotic solution, which is very low in comparison to the procedures described by Sorger et al. [42], which may reduce the influence on the mechanical stability of the bone allograft to a minimum. Additionally, a 10-min impregnation period of allogenic bone grafts in antibiotic solution prior to the surgery has proved to be efficient for proper delivery and it is, therefore, suitable for the surgical procedure [29]. Local administration of antibiotics delivered from cement was introduced in orthopedic surgeries in 1970 [44]. Nowadays, a local administration using spacers or materials, which are impregnated with antibiotics is combined with a systemic antibiotic treatment to improve the outcome of the antimicrobial therapy [25]. Antibiotic-supplemented impacted bone grafts can be used to improve the outcome in revision surgery of infected endoprostheses since systemic applied antibiotics often do not reach sufficient concentrations around the grafts [45,46,47]. Impacted morselized allograft bone is a recognized method to obtain additional support for arthroplasty in revision surgery [46,48,49].

Arthroplastic revision surgeries without the use of cements but including augmentation with bone grafts, improve the bone stock and might be beneficial for the longevity of the implant. In addition, it can be of advantage in the need for further revision surgeries. For this purpose, cancellous bone grafts were reported as antibiotic delivery systems and to augment bone defects [50,51]. A good restoration of the bone stock and low infection rate after revision of total hip replacements was shown for vancomycin-loaded impacted bone allograft [45,47] since in vitro studies have shown the ability of bone grafts to deliver antibiotics [35,46,52,53,54].

Allografts such as the BSM analysed in the present study, which consists not only of the preserved bone matrix based on hydroxyapatite but also of large parts of the collagen matrix, which is preserved by the special purification process without sintering, are of special interest for the combination with antibiotics. In this context, it has been described that collagen promotes antibiotic binding in the case of different other medical devices such as vascular grafts [55,56,57].

Thus, the present study was conducted to analyse the efficacy of human bone allografts prepared according to the C+TBA processing (Cell+Tissuebank Austria GmbH, Krems, Austria) to act as an antibiotic carrier.

The release kinetics of the antibiotics from the loaded allografts were initially examined in vitro directly after immersion, and also in vivo after 1, 3, and 120 days of implantation in the tibia of rabbits. In vitro, the release of clindamycin was above the MIC90 concentration only up to 3 days, while the concentration of gentamicin was under its MIC90 concentration on day 6. In contrast, the concentrations of both vancomycin and the mixture of vancomycin and rifampicin were above their MIC90 levels over the whole observation period of 7 days. Interestingly, the in vivo release measurements showed that the concentrations of clindamycin, vancomycin and the mixture of vancomycin and rifampicin remained above their MIC90 levels up to 3 days post implantation. In contrast, the in vivo concentration of gentamicin fell below its MIC90 level on day 1 post-implantation. Moreover, no further antibiotic loading was found at 120 days post implantation in all study groups.

It has to be mentioned that the spongious C+TBA bone with its network of interconnective pores has a large surface area for the absorption of antibiotics. Thereby, the absorption capacity is not only determined by the large surface due to existing pores, but also by the solubility of the antibiotics and the capacity to bind to proteins, as the C+TBA bone block has a collagen proportion up to 30% in contrast to different other allogeneic but also xenogeneic graft materials [37]. Thus, it is assumable that the protein binding capacity has also an important influence on the impregnation capacity of the allogenic bone blocks. This theory is supported by the individual protein binding capacity of each antibiotic (94% for clindamycin, 46% for vancomycin, <15% for gentamicin, and 87–97% for rifampicin), which is lowest for gentamicin and highest for clindamycin [58,59,60,61]. Nevertheless, other characteristics of the bone blocks than their protein binding capacity seem to play an important role due to a lower initial load of vancomycin compared to gentamicin despite a higher protein binding capacity. Moreover, the binding capacity does not determine the therapeutic efficiency, because it is important to reach the minimal inhibitory concentration (MIC) to kill bacterial pathogens [62,63]. Considering the findings by Holmes et al., who analysed the MIC for 240 MRSA strains, all impregnated bone blocks allowed the delivery of concentrations above the specific MIC90 (8 µg/mL for clindamycin, 0.75 µg/mL for vancomycin, 1 µg/mL for gentamicin and 0.25 µg/mL for rifampicin) after 7 days in vitro and one day and three days in vivo (except for gentamicin) [64]. Pathogens, which often cause implant-associated infections, are usually *Staphylococci* strains and therefore an antibiotic concentration above the MIC90 for MRSA is very important for therapeutic success, especially considering the development of resistances [62,63].

The release measurements clearly demonstrated that the allogeneic bone blocks allow the delivery of high concentrations of antibiotics already within the first days after surgical treatment. In this context, it must be reconsidered what the local release of antibiotics is intended for. Chronic osteomyelitis is a relatively common infection that is normally treated with 4–6 weeks of systemic antibiotics after debridement surgery. However, this time period of antibiotic treatment has no documented superiority over other time intervals, and there is no evidence that prolonged parenteral antibiotics will penetrate the necrotic bone. The local antibiotic application to prevent bone infections was simultaneously included in joint arthroplasty in Europe with the development of this technique in the 1970s. Penicillin, erythromycin, and gentamicin were combined with cement for implant fixation and were detected in high concentrations for extended time periods within the implantation beds [44]. Moreover, Klemm created beads based on commercially available bone cement that are used to manage large dead-space defects pre-impregnated with gentamicin in 1979. The usability of this technique was intended to fill the “dead” space after debridement of infected bone has shown to allow for a cure rate of 91.4% [65,66]. Thereby, the antibiotic-loaded beads should impregnate the hematoma with high levels of antibiotics. In this context, it has been revealed that bioabsorbable ceramic carriers are able to elute therapeutic concentrations of anti-biotic for >7 days [31,32,33,34]. Altogether, it has been shown that such implants show a rapid release of the antibiotic in a more or less controlled manner [67]. The release kinetic measurements in the present study also showed that the majority of antibiotics combined with the allogeneic bone blocks were released after one and three days. This release kinetics would enhance the availability of antibiotics in the wound surroundings during the initial days after implantation, support a systemic therapy and enhance the therapeutic outcome. Thus, it can be concluded that this approach may be rather indicated for prophylaxis of osteomyelitis induced during surgery than for its treatment [68]. Nevertheless, a steady release for a longer period of time would be preferable to exceeding the MIC for a longer period of time [62,63]. Therefore, the release kinetics could be modified to the former binding of antibiotics to the bone substitutes, e.g., by using carrier systems or stronger protein bindings for a prolonged release of coating technologies that allow for a stepwise and controlled release [69,70,71].

Moreover, these results show that the in vitro and in vivo results were comparable in the groups of clindamycin, of vancomycin, and the mixture of vancomycin and rifampicin, while a difference was found in the release kinetics of gentamicin. In the case of this antibiotic agent the results of both study parts differed significantly. These observations suggest that in vitro and in vivo results are not always in concert with each other, and sometimes in vitro testing is not a sufficient precursor to predict the efficiency of a biomaterial or a release kinetic in a dynamic multisystemic micromilieu. Although different multicellular models and cell-matrix-models as well as bioreactors have been developed in the last decades, no completely transferable in vitro model has been developed until now [72]. Hopes are now focused on novel in vitro tissue models based on 3D-printed tissues [73]. Recent studies will show whether these data are completely transferable to more specific questions such as the one investigated here and whether animal testing can be broadly avoided in the future.

Additionally, the resistance tests against *S. epidermidis* ATCC 12228, *S. aureus* ATCC 29213, clinical isolate of *S. epidermidis*, and clinical isolate of MRSA on the four different antibiotics used for loading the allogenic bone grafts were additionally carried out. In comparison to all antibiotics, clindamycin showed the least activity against all tested bacterial strains and fell below the MIC90 after the third day of incubation. Gentamicin showed a higher activity against *S. epidermidis* ATCC 12228, *S. aureus* ATCC 29213 and MRSA compared to vancomycin and clindamycin. Released gentamicin showed the weakest activity against *S. epidermidis* clinical isolate. The released amount of vancomycin showed a similar activity against *S. epidermidis* ATCC 12228, *S. aureus* 29213 and *S. epidermidis* clinical isolate, but a lower efficiency against MRSA. Vancomycin and rifampicin showed a high activity against *S. epidermidis* ATCC 12228, *S. aureus* ATCC 29213 and MRSA but a medium activity against the clinical isolate of *S. epidermidis*, similar to vancomycin. The mixture of vancomycin and rifampicin showed the best results against standard and patient-isolated strains in vitro. This was also observed in the release tests, where clindamycin dropped below the respective MIC90 in vitro and in vivo. Comparable results were observed with gentamicin, even though its concentration remained above the respective MIC90 after 6 days. All investigated bacterial strains exhibited high susceptibility to vancomycin and vancomycin/rifampicin, suggesting that these two groups are sufficient for the combination with the allogenic bone blocks to prevent infections caused by bacterial strains that can enter the implantation site from the patient’s skin or from the hospital’s environment. Interestingly, the strains remained sensitive to these two groups up until 7 days and when combined with the release kinetics tests, this suggests that vancomycin and vancomycin/rifampicin loading of the bone blocks has the potential of preventing the formation of biofilms. This result furthermore supports the assumption that this treatment option can optimally serve as a prophylactic approach.

Histologically, antibiotics were visible at the first two timepoints supporting the aforementioned measurements but were not observed at 120 days post implantation. Moreover, the allografts were fully integrated within the implantation beds with a well-vascularized connective tissue supporting new bone growth. Furthermore, the histomorphometrical analysis of the new bone growth exhibited no significant differences between the different antibiotic groups and non-antibiotic-loaded allograft, which shows that the antibiotic loading does not influence the osteoconductive capacity of the allografts. Interestingly, the allogenic bone grafts induced a low inflammatory host tissue response that is expected to be induced by nearly all bone grafts–also as a sign of its resorption in the course of creeping substitution [74]. The implants induced immune cells (i.e., macrophages and lymphocytes) to the site, as well as biomaterial-associated multi-nucleated giant cells (BMGCs), at later timepoints beside local bone cells such as osteoblasts and osteoclasts. Based on further results the occurrence of these cells (especially of macrophages and BMGCs) suggests that the inflammatory tissue response is an ongoing process that continuously supports vascularization and bone tissue regeneration [75,76,77].

The limitations of this study are related to the measurement of antibiotic concentrations using bioassay, which does not accurately reproduce the tissue environment in the body.

## 5. Conclusions

Altogether, the results of the present study show that the allogenic C+TBA blocks are suitable for releasing high amounts of antibiotics and deliver initial concentrations many times higher than the MIC90 for relevant bacterial strains without delaying bone formation. Due to a fast release of the antibiotic within the first days after implantation, high concentrations can be achieved in the surrounding tissue. This approach ensures a high availability of antibiotics in the affected tissues and can reduce side effects due to the systemic application of antibiotics in high concentrations. Therefore, loading allogenic bone substitutes with antibiotics in combination with a systemic antimicrobial therapy should be further investigated as a strategy for the therapy of implant infections with a high amount of bone loss.

## Figures and Tables

**Figure 1 antibiotics-11-00969-f001:**
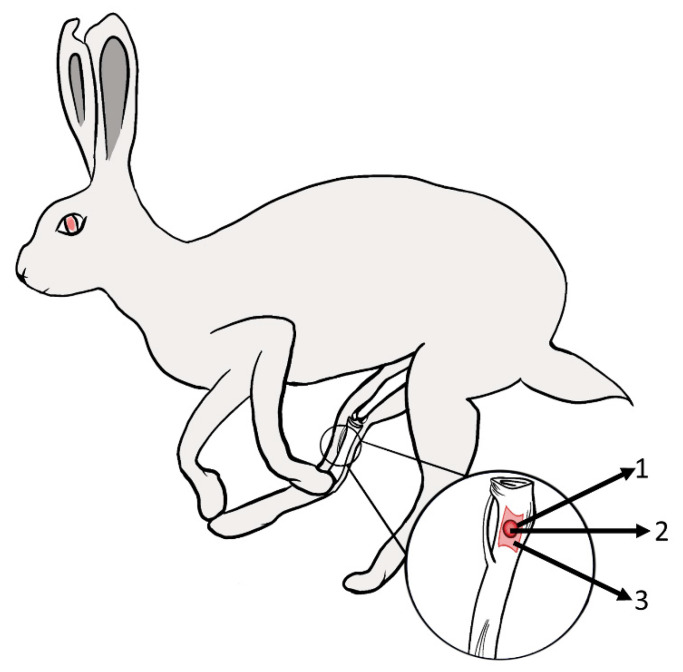
Schematic representation of the rabbit tibia implantation model with a critical-size defect. (**1**) Defect, (**2**) bone graft, and (**3**) barrier membrane.

**Figure 2 antibiotics-11-00969-f002:**
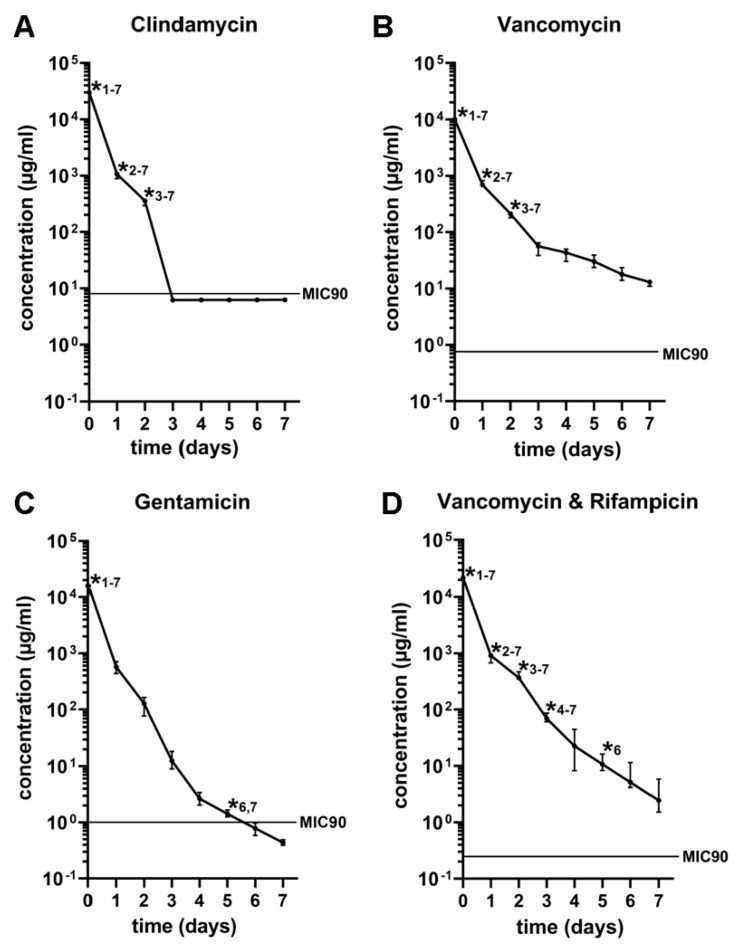
In vitro release kinetics of allogenic bone blocks loaded with antibiotics up to seven days after incubation in clindamycin (**A**), vancomycin (**B**), gentamicin (**C**), and vancomycin in combination with rifampicin (**D**). The concentration of vancomycin and vancomycin in combination with rifampicin, gentamicin, and clindamycin in the bone blocks was above the MIC90 for seven, five, and two days respectively. Significant differences (*p* ≤ 0.05) compared to other time points are marked with a star (*) and the associated time point.

**Figure 3 antibiotics-11-00969-f003:**
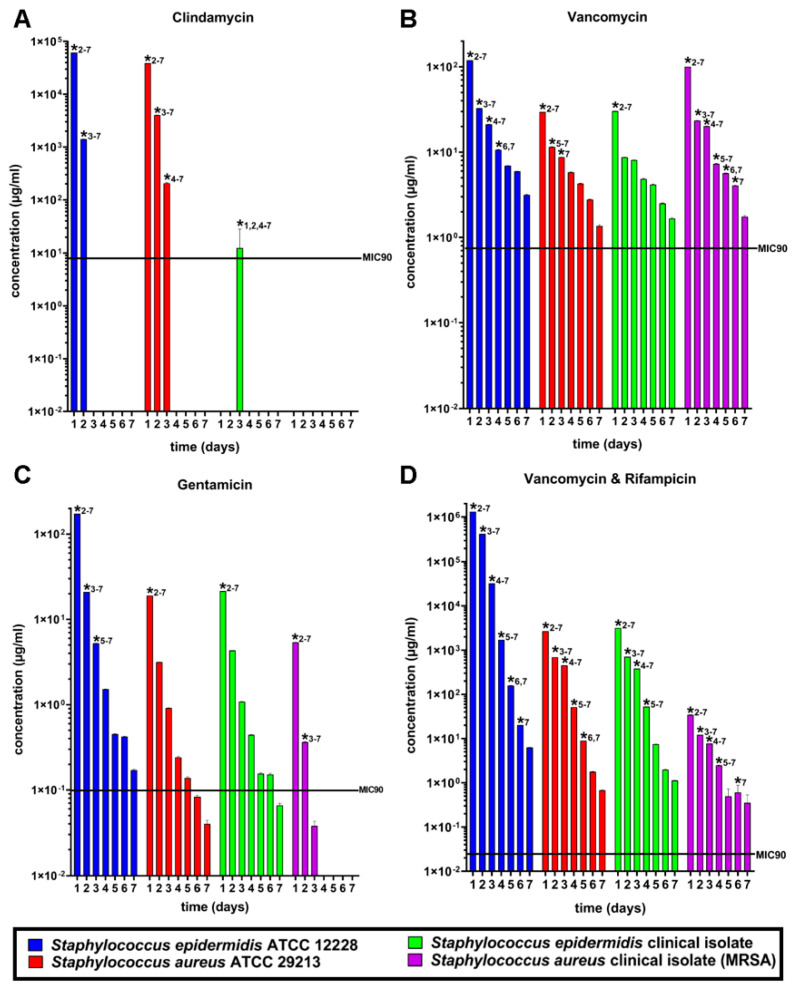
Antibiotic susceptibility tests for different bacterial species and strains. Four antimicrobial substances clindamycin (**A**), vancomycin (**B**), gentamycin (**C**), and vancomycin in combination with rifampicin (**D**) were tested against *Staphylococcus epidermidis* ATCC 12228 (blue), *Staphylococcus aureus* ATCC 29213 (red), a PJI patient isolate of *Staphylococcus epidermidis* (green) and MRSA (purple). Vancomycin and the combination of vancomycin and rifampicin showed the most efficiency, whereas clindamycin and gentamycin showed a reduced efficiency over the time period of seven days. Significant differences (*p* ≤ 0.05) compared to other time points are marked with a star (*) and the associated time point.

**Figure 4 antibiotics-11-00969-f004:**
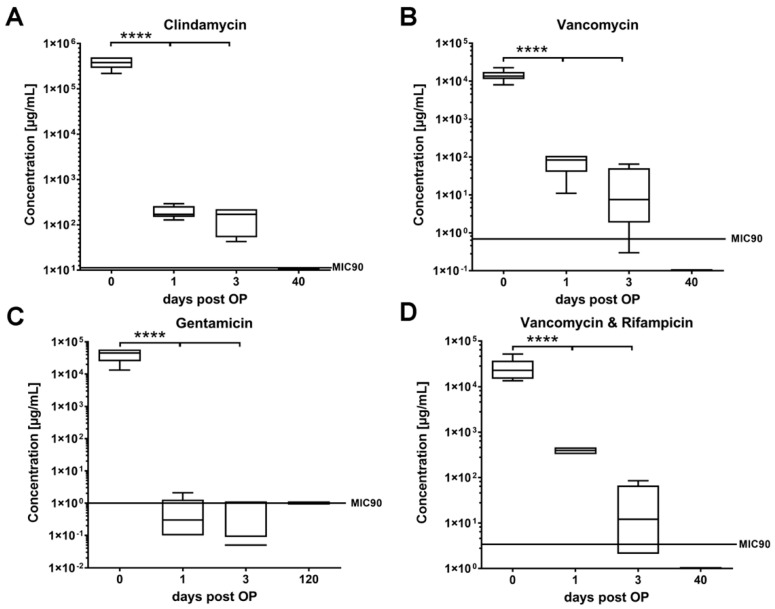
In vivo release kinetics of allogenic bone blocks at 1-, 3-, and 120-days post-implantation. Clindamycin (**A**), vancomycin (**B**), gentamycin (**C**), and vancomycin in combination with rifampicin (**D**). Significant differences (*p* ≤ 0.0001) between initial values and other time points (1 and 3 days) are marked with four stars (****).

**Figure 5 antibiotics-11-00969-f005:**
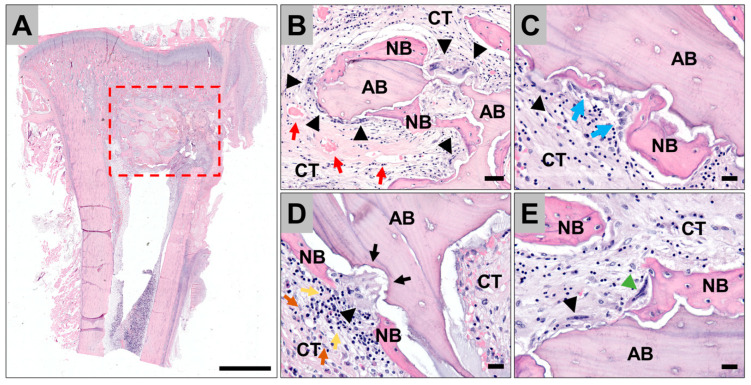
Illustrative histological images of allogeneic bone blocks impregnated with vancomycin implanted in rabbit tibia at 120 days post implantation. Overview of the implantation bed of the allogeneic bone block (red box) (**A**). Closeups of the allogenic bone graft (AB) and new bone growth (NB), surrounded by connective tissue (CT), vessels (red arrows), osteoblasts (blue arrows), osteoclasts (green arrowheads), multinucleated giant cells (MNGCs) (black arrowheads) with visible resorption lacunae (black arrows), lymphocytes (yellow arrows), and macrophages (orange arrows) (**B**–**E**). H&E-stainings, (**A**) 100× magnification, scalebar = 250 µm, (**B**) 20× magnification, scalebar = 50 µm, and (**C**,**E**) 40× magnification, scalebars = 20 µm.

**Figure 6 antibiotics-11-00969-f006:**
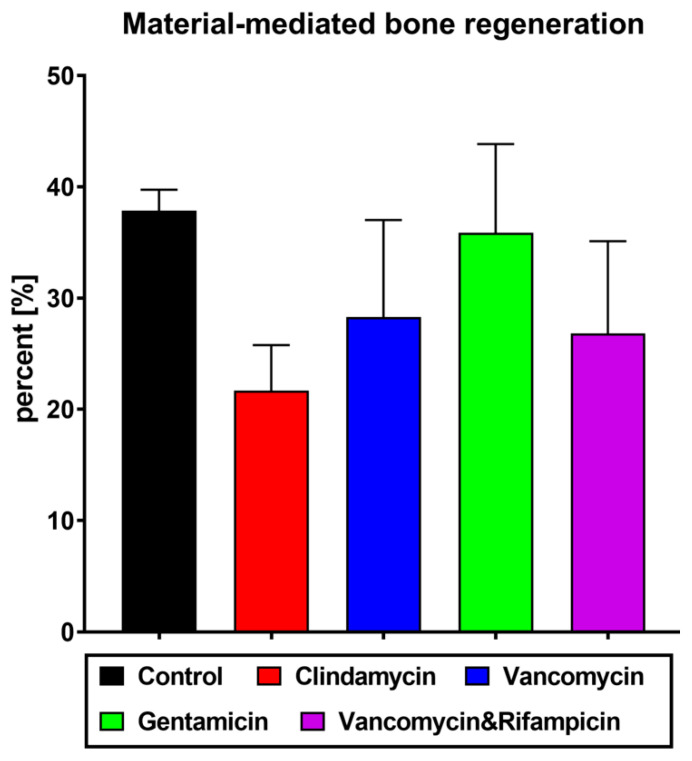
Percentages of bone formation in the different study groups at 120 days post implantation. New bone formation was calculated in relation to the total implant area (in %, displayed as mean ± standard deviation).

**Table 1 antibiotics-11-00969-t001:** *p* values of the statistical comparison.

Study Groups	*p* Values
Vancomycin vs. Clindamycin	0.6871
Vancomycin vs. Gentamycin	0.6316
Vancomycin vs. Vancomycin&Rifampicin	0.9986
Vancomycin vs. Control	0.4277
Clindamycin vs. Gentamycin	0.0960
Clindamycin vs. Vancomycin&Rifampicin	0.8393
Clindamycin vs. Control	0.0607
Gentamycin vs. Vancomycin&Rifampicin	0.4777
Gentamycin vs. Control	0.9954
Vancomycin&Rifampicin vs. Control	0.3019

## Data Availability

All data are presented in the publication.

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
