# Peer review of "Lyophilized Human Bone Allograft as an Antibiotic Carrier: An In Vitro and In Vivo Study"

_antibiotics, 2022, doi:10.3390/antibiotics11070969_

Round 1
Reviewer 1 Report
Comments for “Lyophilized Human Bone Allograft as an Antibiotic Carrier: an in Vitro and in Vivo Study”
In general, authors provided a interesting work in antibiotic field, there are some concerns should addressed:
1. For Figure 2A. the concentration become the same after day 3, this result is quite different with other antibodies, authors should explain this.
2. For Figure 5, it is unclear which antibodies were used in this section
3. For Figure 6, the results suggest antibodies effect the bone reformation, authors should explain this.
Author Response
In general, authors provided a interesting work in antibiotic field, there are some concerns should addressed:
- For Figure 2A. the concentration become the same after day 3, this result is quite different with other antibodies, authors should explain this.
The delivery from the clindamycin from the bone samples showed a standard kinetic after the 3th day. In this study we did not worked with antibodies.
- For Figure 5, it is unclear which antibodies were used in this section
Dear colleague, many thanks for the useful comment. As already described in the Figure legend, Figure 5 includes H&E-stainings and no antibody / immunohistochemical stanings. However, the antibiotic was added.
- For Figure 6, the results suggest antibodies effect the bone reformation, authors should explain this.
Dear colleague, many thanks for the useful comment. As there are no statistical differences between the study groups we decided not to discuss this issue.

Reviewer 2 Report
The work “Lyophilized Human Bone Allograft as an Antibiotic Carrier: an in Vitro and in Vivo Study” by Débora C. Coraça-Huber et al aims to evaluate the efficacy of human bone allografts prepared by C+TBA processing to be used as an antibiotic carrier.
i) In the introduction section:
1. Rows 47-55. I do not understand what the authors want to point out. Either the BSM or allographs? I suggest to rewrite this paragraph.
2. Rows 78-80 “An essential requirement for this form of therapy is a biomaterial, which incorporates large amounts of antibiotics and delivers these substances in a delayed and sustained fashion”. I thin that this phrase should be eliminated, as all the info here, can be found above the phrase.
ii) Materials and Methodd
I would like to suggest to have a subchapter “’Materials”, in which to collect all materials used for your experiments and the suppliers. You only did this for the antibiotics.
iii) Results
1. How many allogenic bone blocks were used for each loading/release experiment?
2. Can you calculate the quantity of the antibiotic on each allograph, before the implantation?
iv) Discussion:
In this paragraph the authors claim that “Allografts such as the BSM analysed in the present study, which consists not only of the preserved bone matrix based on hydroxyapatite but also of large parts of the collagen matrix, which is preserved by the special purification process without sintering”.
Have you used a method (in this study or previous ones) to determine if the bone matrix preserved only hydroxyapatite and collagen? If so, you cand add some details in this paper.
Author Response
In the introduction section:
- Rows 47-55. I do not understand what the authors want to point out. Either the BSM or allographs? I suggest to rewrite this paragraph.
BSM indicates all the substances, which can be used to fill bone voids after reconstructive surgeries, including synthetic and natural substances, like bone allographs. That is the reason we initiated the sentence with this information.
- Rows 78-80 “An essential requirement for this form of therapy is a biomaterial, which incorporates large amounts of antibiotics and delivers these substances in a delayed and sustained fashion”. I thin that this phrase should be eliminated, as all the info here, can be found above the phrase.
Dear colleague, many thanks for the useful comment. We revised the manuscript based on your comment.
Materials and Methods
I would like to suggest to have a subchapter “’Materials”, in which to collect all materials used for your experiments and the suppliers. You only did this for the antibiotics.
Dear colleague, many thanks for the useful comment. The data is already included within the paragraphs 2.1 and 2.2.
Results
- How many allogenic bone blocks were used for each loading/release experiment?
Dear colleague, for the ex vivo/in vitro experiments 3 blocks were used for each experiment and the the experiments we carried out in triplicates. This information was added in the text. For the in vivo experiments 64 bone blocks were used.
- Can you calculate the quantity of the antibiotic on each allograph, before the implantation?
Yes. This information can be found in the Materials and Methods, section 2.2.
Discussion:
In this paragraph the authors claim that “Allografts such as the BSM analysed in the present study, which consists not only of the preserved bone matrix based on hydroxyapatite but also of large parts of the collagen matrix, which is preserved by the special purification process without sintering”.
Have you used a method (in this study or previous ones) to determine if the bone matrix preserved only hydroxyapatite and collagen? If so, you cand add some details in this paper.
Dear colleague, many thanks for the useful comment. We did not conduct such analyses so that we are not able to include this data.

Reviewer 3 Report
The manuscript is focused on the topic of local delivery of antibiotics at the bone surgical sites. The topic is important and interesting; the manuscript is good organized and well written. I have only some minor concerns before the manuscript can be accepted for publication.
L187-188 Why did authors use a collagen barrier membrane? Is it just to get a compartment or hold the allografts in place? Couldn´t they be inserted pressfit? Please add some info in the text.
L190 Authors mention the insertion of one scaffold on each left and right tibia. Were they both of the same antibiotic group in one animal? If not, can it be guaranteed the absence of systemic effects of the antibiotics? The blood level of antibiotics could be analyzed as well. Please add some info in the text.
L207 How were bone allografts removed after 120 days of insertion? Please add some info in the text.
Figure 3. Maybe a more uniform y axis presentation would improve comparability between the groups.
L342. How the antibiotics were detected inside the trabecular structure? Please add some info in the text.
L494, l545, Figure 6: One can see in figure 6 lower bone formation especially for clindamycin. Use as prophylaxis must be weighed against the risk of pseudarthrosis, local side effects, allergic reactions etc.
Some typos: l436, l 487, l504
In order to make it more informative for broader audience of reader (especially for younger scientists) it could be recommended to add (as supplementary material) representative images showing how the analysis of the antibiotics release from implants was performed on agar plates.
Author Response
- L187-188 Why did authors use a collagen barrier membrane? Is it just to get a compartment or hold the allografts in place? Couldn´t they be inserted pressfit? Please add some info in the text.
Dear colleague, many thanks for the useful comment. We revised the manuscript based on your comment.
- L190 Authors mention the insertion of one scaffold on each left and right tibia. Were they both of the same antibiotic group in one animal? If not, can it be guaranteed the absence of systemic effects of the antibiotics? The blood level of antibiotics could be analyzed as well. Please add some info in the text.
Dear colleague, many thanks for the useful comment. The bone blocks that were implanted in one animal were from the same group respectively.
- L207 How were bone allografts removed after 120 days of insertion? Please add some info in the text.
Dear colleague, many thanks for the useful comment. We revised the manuscript based on your comment.
- Figure 3. Maybe a more uniform y axis presentation would improve comparability between the groups.
Even in combination with the graphical presentation of the values with the respective MIC90 concentrations we decided not to uniform the y axes.
- How the antibiotics were detected inside the trabecular structure? Please add some info in the text.
No specific tests were carried out to differenciate the areas in the bone allografts where the antibiotic was loaded. Therefore, the complete information on how the antibiotic detection was realized is explained in the Materials and Methods, section 2.4.1.
- L494, l545, Figure 6: One can see in figure 6 lower bone formation especially for clindamycin. Use as prophylaxis must be weighed against the risk of pseudarthrosis, local side effects, allergic reactions etc.
Dear colleague, many thanks for the useful comment. As there are no statistical differences between the study groups we decided not to discuss this issue.
- Some typos: l436, l 487, l504
Dear colleague, many thanks for the useful comment. We revised the manuscript based on your comment.
- In order to make it more informative for broader audience of reader (especially for younger scientists) it could be recommended to add (as supplementary material) representative images showing how the analysis of the antibiotics release from implants was performed on agar plates.
Unfortunately, we do not have another information to add as supplementary data. However, the description on the Materials and Methods section is clear and the readers can contact us any time for further explanations.

Round 2
Reviewer 1 Report
Authors have addreased my early concerns